# High-Density Microarray Analysis of Microbial Community Structures in Membrane Bioreactor at Short Sludge Retention Time

**DOI:** 10.3390/membranes13020146

**Published:** 2023-01-23

**Authors:** Shilong Li, Liang Duan, Yonghui Song, Slawomir W. Hermanowicz

**Affiliations:** 1State Key Laboratory of Environmental Criteria and Risk Assessment, Chinese Research Academy of Environmental Sciences, Beijing 100012, China; 2Department of Civil and Environmental Engineering, University of California, Berkeley, CA 94720-1710, USA

**Keywords:** high-density microarray, microbial community, membrane bioreactor, membrane sequencing batch reactor, sludge retention time

## Abstract

Membrane bioreactors (MBR) have become prevalent in wastewater treatment because of their high effluent quality and low sludge generation. Sludge retention time (SRT) is an important parameter in the operation of MBR, and it has a direct effect on the microbial community. In this study, microarrays were used to analyze the microbial communities of three different MBRs at short SRTs. The results showed that MBR at SRT 5 days (CS5) has the highest operational taxonomic units (OTUs) richness, but the lowest diversity and uniformity compared to SRT 3 days at continuous CS3 and the sequencing batch (SS3). *Proteobacteria* were the dominant phylum of three reactors. *Bacteroidetes* were the second dominant phylum in MBRs at the continuous model, instead of *Actinobacteria* at the sequencing model. At the class level, the dominant group of *Proteobacteria* exhibited a remarkable difference between the three MBRs. *γ-Proteobacteria* was the dominant group in CS5 and CS3, while *α-Proteobacteria* was the main group in SS3. The samples from the three MBRs had similar compositions of *α*-, *β*- and *δ-Proteobacteria*. However, *γ-Proteobacteria* showed different community compositions at the order level between the three MBRs. *Enterobacteriales* were the dominant group in CS5 and CS3, while *Pseudomonadales* were the dominant group in SS3. The bacterial community concentration of SRT 5 days was generally higher than that of the other two MBRs. The community composition of CS5 was significantly different from that of CS3 and SS3, and the phylogenetic relationships of the three MBRs were relatively different.

## 1. Introduction

In the field of water treatment, more and more new reactors composed of membranes and bioreactors have been applied [1,2,3,4]. Membrane bioreactor (MBR), which combines membrane separation and the activated sludge process, has become prevalent in wastewater treatment [5,6,7,8]. Due to the introduction of membranes, compared with the traditional activated sludge process, the effluent quality of MBR has been greatly improved, and sludge production has been reduced [9,10], thus, attracting widespread interest. Sludge retention time (SRT) is an important parameter in the operation of MBR, and it is widely assumed that a longer SRT is conducive to the growth of nitrifying bacteria and the improvement of nitrification ability [11]. MBR with short SRT, on the other hand, can also achieve better nitrogen removal [12], which is closely related to the growth of various micro-organisms. Therefore, it is very important to investigate the microbial community structure of MBR at short SRTs.

Microarray technology is a new genomics research method and can be used as a high-throughput method for microbial community analysis [13]. The use of high-density microarrays to analyze the composition and structure of the microbial community in the MBR has unique advantages. Unlike other taxonomic nucleic acid-based analysis methods, which are limited by the speed of sequence analysis, microarrays can simultaneously provide a high-throughput tool for the detection, identification, characterization, and quantification of micro-organisms in natural environments. Microarray technology can enable the quantitative analysis of the microbial community based on fluorescence intensity. OTU abundance can be obtained by measuring fluorescence intensity, and the relative OTU abundance *x* (0 < *x* < 1) can be obtained by the normalization of the fluorescence maximum value [13]. In addition, high-density microarrays have the advantage of classifying and identifying a mass of microbial communities without the need for subsequent DNA isolation and sequencing. In recent years, it has been reported that high-density microarray technology has been used to carry out some research related to environmental systems [12,14,15].

In this study, two MBRs and a membrane-sequencing batch reactor (MSBR) with short SRTs were operated. In order to reveal the differences of microbial community structures in different short SRT MBRs and the MSBR, we used a high-density microarray containing 297,851 probes targeting 8935 clusters of 16S rRNA genes (PhyloChips) to investigate the composition of microbial communities.

## 2. Materials and Methods

### 2.1. Bench-Scale Submerged Membrane Bioreactors (MBR)

Three lab-scale aerobic submerged membrane bioreactors were operated in parallel at SRT 3 days (CS3), 5 days (CS5) and the sequencing batch at SRT 3 days (SS3) to treat synthetic municipal wastewater. Hollow-fiber membranes (Sterapore LHEM03334, Mitsubishi Rayon, Tokyo, Japan), with a total area of 0.03 m^2^, were installed in a 4 L aerobic-activated sludge tank. The material of the membranes was polyethylene, and the separation particle size was 0.4 µm. Continuous aeration was provided underneath the membranes to supply the dissolved oxygen concentration of approximately 9 mg/L and prevent membrane fouling. All MBRs were operated in constant flux mode at the same hydraulic retention time of 6 h. The MSBR (SS3) was operated under the following strategy: filling time (10 min), reaction time (290 min) and draw time (60 min). The target SRTs were maintained through the direct removal of sludge from the bioreactor on a daily basis. Synthetic wastewater, mainly composed of acetate and corn starch as carbon sources, was used to provide a consistent influent feed composition.

### 2.2. Analytical Methods for Water Quality Parameters

The influent and permeate quality as well as the membrane suction pressure were monitored routinely. Total chemical oxygen demand (COD), total nitrogen and ammonia-nitrogen were measured according to the manufacturer’s instructions using Hach Methods 8000, 8039 and 8008, respectively.

### 2.3. Microarray

In this study, we used G2 microarray chips, designed by Lawrence Berkeley National Laboratory and synthesized by Affymetrix Inc. (Santa Clara, CA, USA), to identify the bacterial species. The chips had an approximate density of 10,000 molecules/μm^2^ and included 297,851 probes targeted to 8935 clusters (operational taxonomic units, OTUs) in 16S rRNA gene sequences. The average number of replicated probes chosen for each cluster was 24 to ensure precision.

PhyloChips were scanned and recorded as a pixel image, and individual signal values and intensities were completed using standard Affymetrix software (GeneChip microarray analysis suite, version 5.1). The positive fraction (Pf value) was calculated as the number of positive probe pairs divided by the total number of probe pairs in each probe set. An OTU was considered “present” when its Pf value was greater than 0.9. PhyloChip results are output as lists of detected OTUs and their hybridization scores, along with associated taxonomic information and references to represented sequences in public 16S rRNA gene repositories (greengenes.lbl.gov).

## 3. Results and Discussion

### 3.1. The Process Performance

The three MBR reactors demonstrated excellent performance in removing organic matter and nitrogen. The COD removal rate of the three reactors can reach more than 94%, among which the NH_4_^+^-N removal rate of SS3 can reach more than 85%, and the NH_4_^+^-N removal rate of CS3 and CS5 can reach more than 87%. The COD concentration in the effluent water of the three reactors was less than 40 mg/L, and the NH_4_^+^-N concentration was less than 10 mg/L. This indicates that the MBRs had a good effect on the removal of organic matter and ammonia nitrogen under the condition of short SRT.

### 3.2. The Diversity and Similarity of Microbial Communities

The richness of microbial communities in the three MBRs was examined by a microarray. A total of 1260 OTUs were detected in at least three samples, accounting for 14% of the OTUs on the PhyloChip. All detected OTUs were taxonomically derived from 36 bacterial phyla, 73 classes, 148 orders, 279 families and 351 subfamilies.

The microbial communities in the three reactors showed extremely high diversity, as reflected in the fact that 34 (CS5), 35 (CS3) and 33 (SS3) identified bacterial phyla were detected. As shown in Table 1, the total numbers of OTUs detected by microarrays were 1031 (CS5), 954 (CS3) and 869 (SS3) in the three MBRs, indicating that CS5 had the most richness. It is mainly attributed to the fact that a relatively longer SRT was adopted in CS5. Different operating conditions, such as aeration mode and dissolved oxygen, affect the distribution of microbial communities in bioreactors [16]. It has been reported that longer SRT could promote the development of slow-growing micro-organisms, such as *Nitrospira* and *Comammox* [17]. However, the Shannon–Wiener diversity, Simpson index and equitability index of CS5 were the lowest among the three samples, suggesting that the microbial community diversity of CS5 was lower than the other two MBRs. It could be inferred that the bacterial communities in CS5 were distributed less evenly than the other two MBRs. Therefore, with the increase in SRT from 3 days to 5 days, the species richness of microbial communities increased, but their uniformity decreased.

There were more shared microbial communities and fewer distinct communities in the three reactors (Figure 1). Each of the three reactors had a very small number of unique microbial communities. The numbers of unique microbial communities were 97 (CS5), 96 (CS3) and 82 (SS3), and the number of shared microbial communities in the three groups was 607. The similarities between pairwise reactors in the three reactors were very low and could only reach 46.5% (CS3-CS5), 44.4% (SS3-CS5) and 43.1% (CS3-SS3). Additionally, the similarities between MSBR and MBRs were lower than those between the two MBRs, which may be caused by different operation modes. The dominant community in parts A and B are *γ-Proteobacteria,* accounting for 23%, while the dominant microbe in part C is *α-Proteobacteria* (33%). *γ-Proteobacteria* can be involved in the degradation of nitrate and other contaminants in bioreactors [18,19], and *α-Proteobacteria* also play an important role in the denitrification reaction [20]. Moreover, the three micro-organisms with the highest proportion in D, E, F and G are γ-proteobacteria (15–25%), *β-Proteobacteria* (10–15%) and *α-Proteobacteria* (17–22%).

### 3.3. The Dominant Bacteria

As shown in Figure 2, *Proteobacteria* was the dominant phylum and accounted for 59% to 64% of the total bacterial community. The result was consistent with previous studies on MBRs analyzed by conventional molecular biology methods [21]. This group of *Proteobacteria* has considerable morphological, physiological and metabolic diversity, which are of great importance to global carbon, nitrogen and sulfur cycling [22]. *Bacteroidetes* were detected as the second most prevalent phylum in CS5 (10%) and CS3 (11%). The members of *Bacteroidetes* are widely distributed in the environment and are closely related to the degradation of organic matter and the cycle of nitrogen [23]. *Actinobacteria* (5% for CS5, 8% for CS3) and *Firmicutes* (5% for CS5, 6% for CS3) were the third and fourth most prevalent groups of bacteria detected in this study. The reactor SS3 showed different characteristics of microbial community structure. *Actinobacteria* (9%) was the second dominant bacterial community instead of *Bacteroidetes* in CS5 and CS3. Moreover, it is worth pointing out that the species richness of *Actinobacteria* (74 OTUs) and *Chloroflexi* (23 OTUs) in SS3 was the highest in three MBRs, despite most of the phyla in SS3 were the lowest. It has been reported that MSBR has a higher COD removal rate than MBR [24]. This phenomenon can be explained by the higher proportion and species richness of *Actinobacteria* in MSBR found in this study. The members of the *Actinobacteria* phylum are a group of gram-positive bacteria that have an important role in organic matter turnover and carbon cycling [25]. Other phyla, including *Acidobacteria*, *Chloroflexi*, *Cyanobacteria*, *Cyanobacteria*, *Spirochaetes*, *Verrucomicrobia*, *Planctomycetes*, *Gemmatimonadetes* and unclassified only occupied no more than 3% of the total bacterial community in three MBRs. Therefore, this study provides a comprehensive survey of the richness and composition of MBR microbial communities at different operating conditions.

The class-level identification of the bacterial communities in three reactors is illustrated in Figure 3. At the class level, *α-Proteobacteria*, *β-Proteobacteria* and *γ-Proteobacteria* were the main community groups of *Proteobacteria* in three MBRs. However, the dominant group of *Proteobacteria* exhibited a remarkable difference between the three MBRs. *γ-Proteobacteria* was the dominant group in CS5 and CS3, while *α-Proteobacteria* was the main group in SS3. However, *β-Proteobacteria* was the dominant group reported in a study related to the MSBR [26]. Moreover, the relative abundance of *γ-Proteobacteria* in CS3 (41%) was higher than the other two MBRs. The previous publications have reported that specific phylogenetic groups belonging to *γ-Proteobacteria* are likely the pioneers of surface colonization on membranes, which could lead to severe irreversible membrane biofouling [27]. In addition, SRT is inversely proportional to membrane fouling [28], which explains why CS3 has the highest relative abundance of *γ-Proteobacteria*. For the *Bacteroidetes* phylum, *Sphingobacteria* and *Flavobacteria* were the most dominant classified subgroups, accounting for 31–40% and 29–38% of the total bacterial community in the three MBRs, respectively. Both are inseparable, and they are involved in the nitrogen cycle [29]. For the *Firmicutes* and *Actinobacteria* phylum, *Clostridia* and *Actinomycetales* were the dominant groups.

The order level of *Proteobacteria* was conducted to reveal more information on the microbial community evolution between the three bioreactors (Figure 4). It was obvious that the samples from the three MBRs had similar compositions of *α-*, *β-* and *δ-Proteobacteria*. Within the *α-Proteobacteria*, *Rhizobiales*, *Rhodobacterales* and *Bradyrhizobiales* were the dominant groups within a narrow range of 22–23%, 16–25% and 17–20%, respectively. Within the *β-Proteobacteria*, *Burkholderiales* was the dominant group and constituted 70–81% of the detected OTUs. *Rhodocyclales* was the second dominant group and constituted 8–13%. The seven other detected groups (*Ellin6095/SC-I-39*, *Hydrogenophilales*, *Methylophilales*, *MND1 clone group*, *Neisseriales*, *Nitrosomonadales* and unclassified) had many fewer detections (1–9 OTUs) in all samples and constituted approximately 11–17% of *β-Proteobacteria*. Within the *δ-Proteobacteria*, *Desulfobacterales*, *Desulfovibrionales*, *Myxococcales* and *Syntrophobacterales* were the dominant groups, constituting 21–31%, 14–19%, 17–18% and 17%, respectively. However, *γ-Proteobacteria* showed different community compositions at the order level between the three MBRs. *Enterobacteriales* were the dominant group of CS5 and CS3 (constituting 22% and 24% of the detected OTUs), while *Enterobacteriales* only constituted 5% of the detected *γ-Proteobacteria* in SS3. *Pseudomonadales* was the dominant group in SS3 and occupied 34%, and a few members, such as species of *Pseudomonas*, played a central role in denitrification [30,31]. Bacteria in the *Enterobacteriaceae* family are gram-negative bacteria that are shaped similar to rods and are usually 1–5 μm in length. They are facultative anaerobic bacteria that ferment sugars to produce lactic acid and other products. Most of them can also convert nitrate to nitrite. However, unlike most similar bacteria, some members of the *Enterobacteriaceae* typically lack cytochrome C oxidase. Although there are exceptions, such as *Plesiomonas Shiga* [32]. This indicates that the denitrification intensity of MBR was higher than that of MSBR. The reason for this difference was that the two systems were operated in different modes. It has been previously reported that the continuous system (MBR) had higher bacterial community diversity than the discontinuous system (MSBR) [33]. This result also corresponds to the removal efficiency of ammonia nitrogen in Section 3.1.

### 3.4. The Concentration and Phylogenetic Analysis of Microbial Communities

As shown in Figure 5, the average fluorescence intensity of micro-organisms at the phylum level in CS5 was higher than that of CS3 and SS3. This indicated that the microbial concentration in the reactor with an SRT of 5 days was higher than that in the reactor with an SRT of 3 days. Except for *α-*, *β-*, *γ*- and *ε-Proteobacteria*, the concentration of other microbial communities in CS3 was higher than that in SS3. In the three reactors, the micro-organisms with the highest fluorescence intensity were *β-Proteobacteria* (CS5), *Unclassified*-*Proteobacteria* (CS3) and *ε-Proteobacteria* (SS3), respectively.

As shown in Figure 6, among the top 50 genera with the most abundant communities in the three reactors, green means less abundant, and red means more abundant. The community composition of CS5 was significantly different from that of CS3 and SS3, and the community distribution of CS3 and SS3 is similar at the genus level. This difference could be attributed in large part to the different SRTs (3 and 5 days). Most of the bacteria in CS5 were greatly abundant, and only a few had relatively low abundance, while the opposite was true for CS3 and SS3. *Bacterias* at the genus level with the highest concentrations belong mainly to *Cyanobacteria*, *α*-, *β*- and *γ-Proteobacteria* in CS3 and SS3. To further evaluate the differences in microbial community structure among the three MBRs, phylogenetic relationships of the top 50 most abundant genera of the bacterial community for each sample are shown in Figure 7. The phylogenetic tree showed that the microbial communities of the three samples could be divided into three categories, but the phylogenetic relationships of the three samples were relatively different. Among them, the bacteria with Genbank numbers AJ012698.1 (*Acetobacteraceae*) and AY192273.1 (*Unclassified Alphaproteobacteria*) in CS3 had obvious genetic differences from other members. For the top 50 most abundant genera, CS3 and CS5 were more similar in phylum level classification, while the classification of SS3 was significantly different from that of CS3 and CS5 at the phylum-level, indicating that different operation modes had a significant impact on the distribution of microbial communities.

## 4. Conclusions

The three short SRT MBRs all had satisfactory removal effects for organic matter and NH_4_^+^-N. The removal efficiencies can reach 94% and 85%, respectively. Three MBRs showed high microbial diversity, and CS5 had the highest OTU richness but the lowest diversity and uniformity compared to CS3 and SS3.

*Proteobacteria* was the dominant phylum of the three reactors. *Bacteroidetes* was the second dominant phylum in CS5 and CS3 instead of *Actinobacteria* in SS3. At the class level, *γ-Proteobacteria* was the dominant group in CS5 and CS3, while *α-Proteobacteria* was the main group in SS3. The samples from the three MBRs had similar compositions of *α-*, *β-* and *δ-Proteobacteria*. However, *γ-Proteobacteria* showed different community compositions at the order level between the three MBRs. *Enterobacteriales* were the dominant group of CS5 and CS3, while *Pseudomonadales* were the dominant group of SS3. This indicates that the denitrification intensity of MBR was higher than that of MSBR, which was caused by different operating modes.

The bacterial community concentration of CS5 was generally higher than that of CS3 and SS3. The community composition of CS5 was significantly different from that of CS3 and SS3, and the community distribution of CS3 and SS3 was similar at the genus level. The microbial communities of the top 50 most abundant genera for each sample could be divided into three categories, but the phylogenetic relationships of the three samples were relatively different.

## Figures and Tables

**Figure 1 membranes-13-00146-f001:**
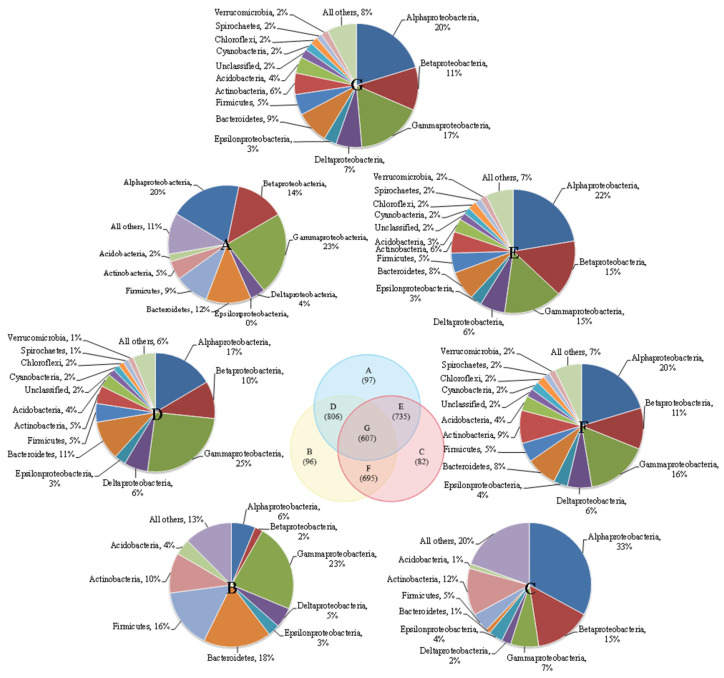
Venn diagram of three MBRs communities overlapped.

**Figure 2 membranes-13-00146-f002:**
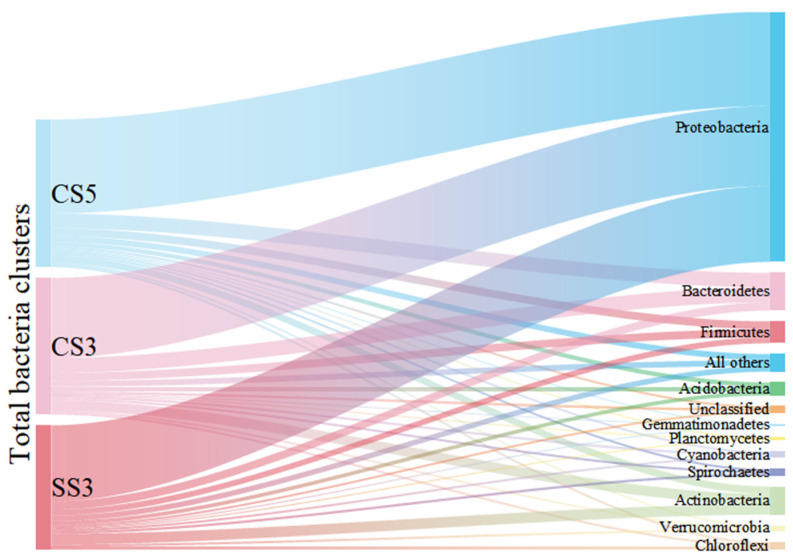
Sankey diagram of taxonomic breakdown (at the phylum level) of three MBRs.

**Figure 3 membranes-13-00146-f003:**
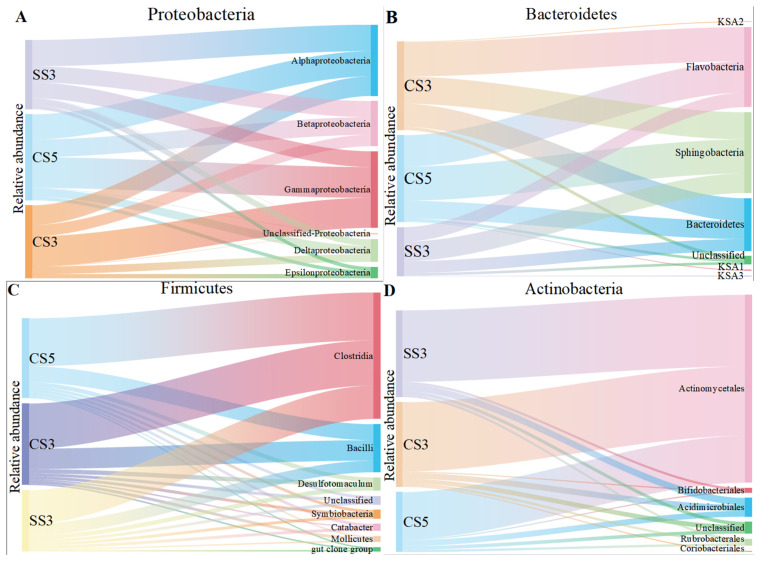
The taxonomic distribution of the *Proteobacteria* (**A**), *Bacteroidetes* (**B**), *Firmicutes* (**C**) and *Actinobacteria* (**D**) classes for the microarray.

**Figure 4 membranes-13-00146-f004:**
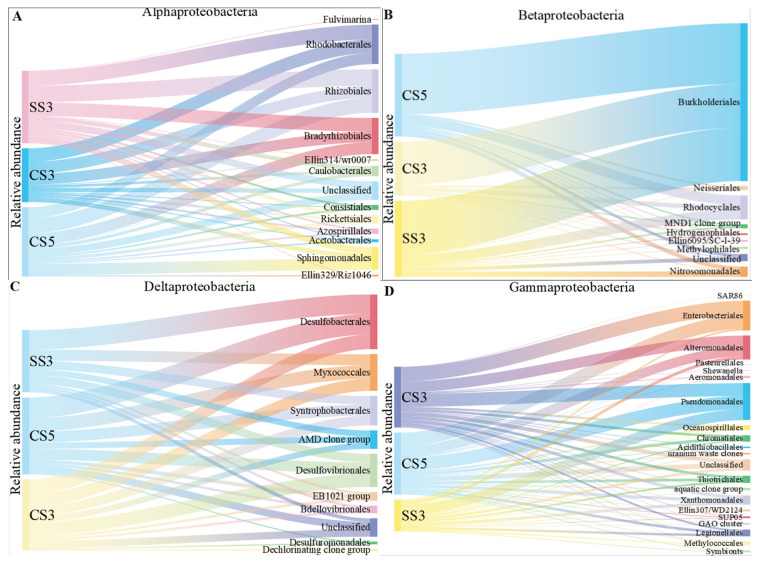
The taxonomic distribution of the *α-Proteobacterial* (**A**), *β-Proteobacterial* (**B**), *δ-Proteobacterial* (**C**) and *γ-Proteobacterial* (**D**) orders on the microarray.

**Figure 5 membranes-13-00146-f005:**
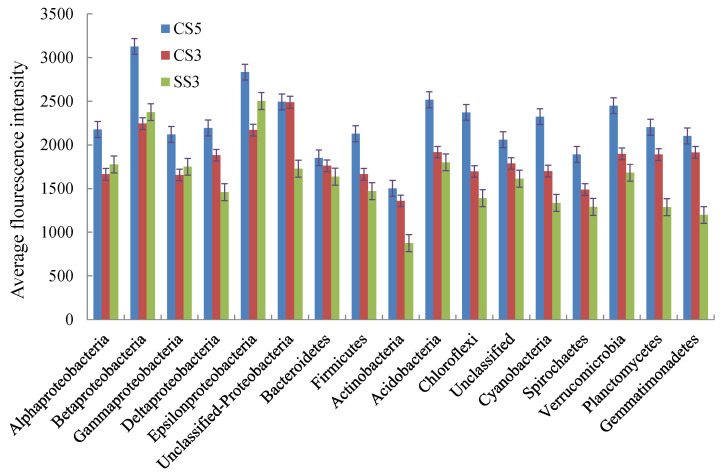
Average PhyloChip hybridization signal intensities for the three MBRs at the phylum level.

**Figure 6 membranes-13-00146-f006:**
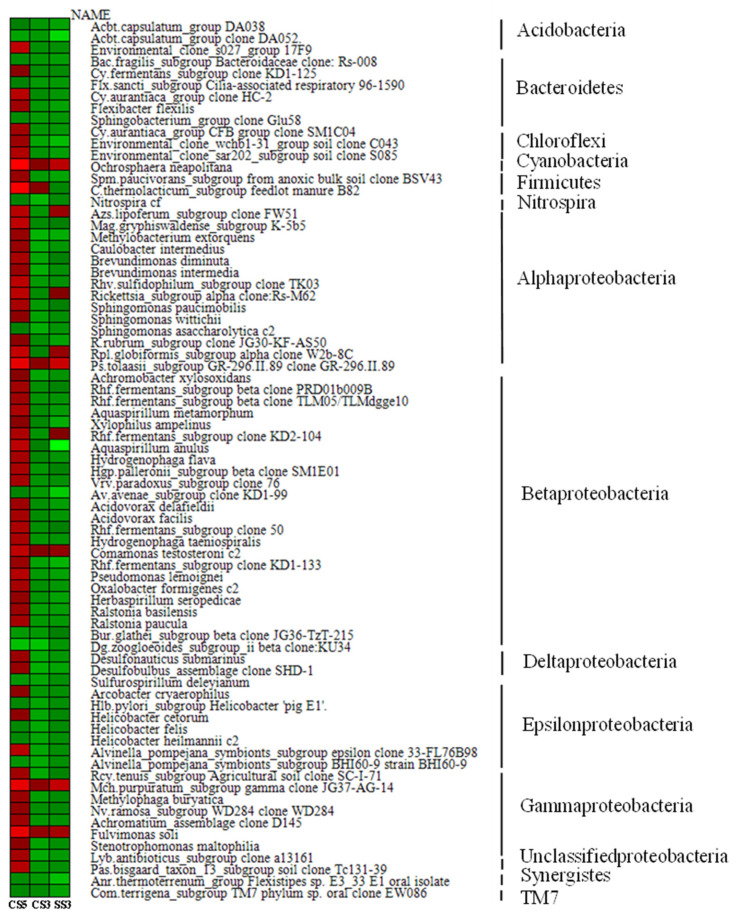
Heatmap showing the top 50 most abundant genera of bacterial community for each sample.

**Figure 7 membranes-13-00146-f007:**
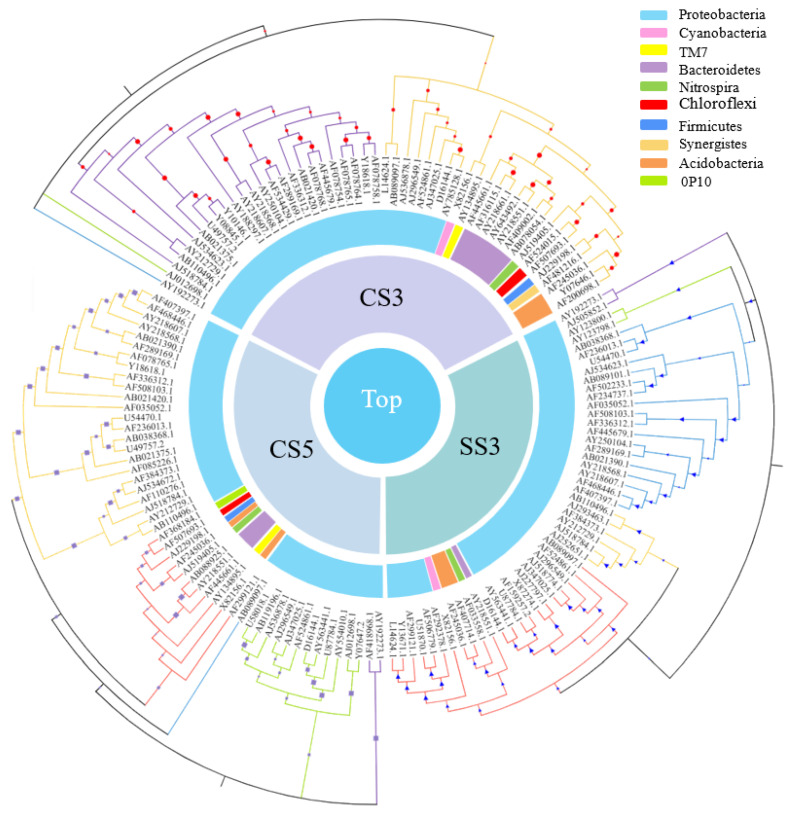
Phylogenetic relationships of the top 50 most abundant genera of the bacterial community for each sample.

**Table 1 membranes-13-00146-t001:** Estimates of relative diversity based on the 16S rDNA PhyloChip analysis.

	CS5	CS3	SS3
**Species richness(S)**	1031	954	869
**Shannon–Wiener Diversity (H’)**	1.57	1.69	1.68
**Simpson index**	0.58	0.63	0.61
**Equitability index (EI)**	0.23	0.25	0.25

## Data Availability

The data presented in this study are available upon request from the corresponding author.

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
