# Peer review of "High-Density Microarray Analysis of Microbial Community Structures in Membrane Bioreactor at Short Sludge Retention Time"

_membranes, 2023, doi:10.3390/membranes13020146_

Round 1

Reviewer 1 Report

This manuscript investigates the microbial community in response to different SRTs and the changes was revealed, yielding some instructions for the MBR application. Overall, the work is interesting. However, I have some major concerns that should be addressed before this paper can be considered for publication.   1)      Line 18-21, Abstract, genus rather than phylum was more meaningful. In addition, the abstract should cover more information in the manuscript. 2)      Section 3.1, the performance of MBR is very important, thus the data should be given. 3)      Section 3.4, wrong title, kindly suggest revising the sentence. 4)      With regards to SRT and microbial community, kindly suggest indicating the significant value of the parameters shaping the microbial community. Which parameter is significant? (Sci Total Environ 2022, 853, 158424). 5)      Line 197-205, kindly explain more about the Fig.6 and Fig.7. 6)      Conclusion can be more informative.  

Reviewer 2 Report

Comments to membranes-2156009: In this article, microarray was used to analyze the microbial communities of three different MBRs at short SRT. Proteobacteria was the dominant phylum of three reactors. Bacteroidetes was the second dominant phylum in MBR at continuous model instead of Actinobacteria in MBR at sequencing model. In general, the article is interesting and the results are substantial. However, I have several concerns about this article:

1.     All abbreviations should be defined when they first appear, but there is no need to repeat the definition, such as MBR.

2.     The introduction part should be revised to highlight the topic of high-density microarray analysis of microbial community structures in MBR.

3.     The unit “m2” in section 2.1 and 2.3 should be “m2”.

4.     I did not find any results or figures on COD and NH4+-N in the main text.

Reviewer 3 Report

The manuscript studied membrane bioreactors in the field of wastewater treatment. A novel sequencing method, microarray, was used to analyze the distribution of microbial communities in membrane bioreactors with short sludge residence time. It is of great significance to the development of MBR. There are still some problems in the paper, it is recommended to accept after modification. Some suggestions are as follows for you to revise your manuscript. 

1 How does the microarray method quantify, by the intensity of the fluorescence? It is suggested to add relevant explanation in the introduction section.

2 Some dominant microorganisms are listed in this paper, and their functions and roles in wastewater treatment should be elaborated

3 You can improve “Figure 7. Phylogenetic relationships of the top 50 most abundant genera of bacterial community for each sample.” by referring to the latest literature with in recent 3 years.

4 The references related to microorganisms are too few, so the references should be further supplemented according to the content of the paper.

5 The sentence in lines 180-181 requires clarification and reference to the source.

Round 2

Reviewer 1 Report

The comments raised by the reviewer have been addressed, thus can be accepted for possible publication.